# Economic Burden of SARS-CoV-2 Patients with Multi-Morbidity: A Systematic Review Protocol

**DOI:** 10.3390/ijerph192013157

**Published:** 2022-10-13

**Authors:** Amirah Azzeri, Mohd Noor Afiq Ramlee, Mohd Iqbal Mohd Noor, Mohd Hafiz Jaafar, Thinni Nurul Rocmah, Maznah Dahlui

**Affiliations:** 1Faculty of Medicine & Health Science, Universiti Sains Islam Malaysia (USIM), Persiaran Ilmu, Bandar Baru Nilai, Nilai 71800, Negeri Sembilan, Malaysia; 2Department of Research Development and Innovation, University of Malaya Medical Centre (UMMC), Lembah Pantai, Kuala Lumpur 59100, Malaysia; 3Faculty of Business Management, Universiti Teknologi MARA (UiTM) (Pahang), Raub 27600, Pahang, Malaysia; 4Institute for Biodiversity and Sustainable Development, Universiti Teknologi MARA (UiTM), Shah Alam 40450, Selangor, Malaysia; 5Department of Health Administration and Policy, Faculty of Public Health, Universitas Airlangga, Surabaya 60115, Jawa Timur, Indonesia; 6Centre of Population Health, Department of Social and Preventive Medicine, Faculty of Medicine, Universiti Malaya, Kuala Lumpur 50603, Malaysia

**Keywords:** COVID-19, economic burden, comorbidity, cardio/cerebrovascular, diabetes, hypertension, respiratory diseases

## Abstract

Economic burden issues in SARS-CoV-2 patients with underlying co-morbidities are enormous resources for patient treatment and management. The uncertainty costs for clinical management render the healthcare system catatonic and incurs deficits in national annual budgets. This article focuses on systematic steps towards selecting and evaluating literature to uncover gaps and ways to help healthcare stakeholders optimize resources in treating and managing COVID-19 patients with multi-morbidity. A systematic review of all COVID-19 treatment procedures with co-morbidities or multi-morbidity for the period from 2019 to 2022 was conducted. The search includes studies describing treatment costs associated with multi- or co-morbidity cases for infected patients and, if concurrently reported, determining recurring expenses. Study selection will follow the Preferred Reporting Items for Systematic Reviews and Meta-Analyses (PRISMA) guidelines. Galbraith plots and I^2^ statistics will be deployed to assess heterogeneity and to identify potential sources. A backward elimination process will be applied in the regression modelling procedure. Based on the number of studies retrieved and their sample size, the subgroup analysis will be stratified on participant disease category, associated total costs, and degree of freedom in cost estimation. These studies were registered in the PROSPERO registry (ID: CRD42022323071).

## 1. Introduction

SARS-CoV-2 infection significantly negatively impacts human lives and has dramatically affected the global economy. During the start of the COVID-19 waves in late 2019, the World G.D.P. (World Gross Domestic Product per capita) growth rate was 2.33% (a decline of 0.7% compared to 2018), which took a significant plunge in 2020, when the total G.D.P. marked its greatest decline at −3.60% (a drop of 5.93% from 2019), the lowest since 1961 [1,2]. Most human activities halted as most countries opted to undergo lockdowns to reduce cases. Due to those restrictions, there was a massive impact on the economy and the environment; there are several reports that indicate the reduced economic consumption of resources, such as electricity [3,4], food [5], and necessities [5,6,7], resulting in financial losses for businesses and vice versa and government revenue in the form of taxes. Despite that, substantial resources were poured into the healthcare system to combat the pandemic. The resilience of the COVID-19 pandemic and the spread of this virus have significantly impacted the healthcare system [8].

As time passes, more mutations of SARS-CoV2 emerge. Although most changes in the COVID-19 variants have little to no impact on the virus properties, some of these alterations threaten global healthcare [9,10]. The iterations of the variant of concern (V.O.C.) introduce a new parameter in the healthcare system, as a new V.O.C. results in different disease severity, is easily spread, or even compromises the performance of vaccines, medicines, diagnosis tools, and other public health measures [9,10,11,12]. The ever-changing nature of some V.O.C., especially VOC-LUM (variant of concern linage under monitoring) cases, is becoming much more difficult for patients who suffer from co-morbidities before contracting the virus as clinical management becomes more elaborate and urgent and increasing mortality based on prior and new complication results. The co-circulation of variants with additional new V.O.C. renders the healthcare system catatonic. At the same time, the existing morbidities of the patient’s treatment regime will differ while using more resources.

These complications open up a new avenue for researchers to identify the resources used in treating patients suffering from the SARS-CoV-2 virus with prior morbidities to further identify the essential resources needed to treat patients and to further strengthen COVID-19 responses while adequately allocating time and healthcare resources required in similar cases [8,13,14,15,16,17]. In order to uncover the complications in COVID-19 patients with multi-morbidities, our research focuses on systematic steps toward selecting and evaluating the scientific literature to find gaps [16,18] and the way forward to help healthcare stakeholders further understand and provide informed decision-making procedures to optimize resources in treating and managing said underlying conditions.

The primary question for this systematic review is, “what is the economic effect of COVID-19 patients that suffer multi-morbidity towards medical practitioners?”. Hence, to answer the research question, this review aims to determine the direct medical costs associated with the clinical management of COVID-19 patients with background co-morbidities or who have multi-morbidity. This research question is broken down into the Population, Intervention, Comparison, and Outcome scheme (PICO). The above primary question methodologies contain critical elements of the keyword’s column [18].

## 2. Materials and Methods

### 2.1. Methodological Overview

This systematic review protocol applies the Guidelines and Standards for Cochrane Reviews under the COVID-END initiative by PROSPERO. The reporting of the full systematic review will follow the PRISMA Statement 2020 for the review protocol, as all research article search databases, registries, and other sources are compiled according to PRISMA Statement 2020, as shown in Figure 1 and Figure 2 [19]. The essential aspects of the review process are written down and kept as a permanent record [20,21]. PROSPERO’s goal is to give a comprehensive listing of systematic reviews registered at the start of the project to avoid duplication and to reduce the risk of reporting bias by comparing the finished review with the protocol’s plans. Reducing the risk of bias will follow the Cochrane Handbook, where the specifications of bias assessment will focus on the study and outcomes of the screened articles [22,23].

### 2.2. Bibliographic Databases

The literature search for the systematic review will be undertaken using Web of Science, Scopus, PubMed, G.H.L., V.H.L., EMBASE, Cochrane, POPLINE, and SIGLE (refer to Figure 1). In the initial step, before embarking on searching the databases for a journal, the authors will initiate a search for issue classification in the Cochrane or a Cochrane-related database (such as PROSPERO) to avoid replication when conducting the literature review (refer Figure 2) [24,25].

In Web of Science, PubMed, G.H.L., V.H.L., and POPLINE, the search strategy will be run based on the “topic” (T.S.) field, which includes literature titles, research abstracts (short and/or extended abstract), keywords, and “Keywords Plus” (automatically generated terms pulled from the labels of cited articles).

In Scopus, EMBASE, SIGLE, and MEDLINE, literature titles, research abstracts, and keywords will be searched using the exact search string used in the Web of Science database, outlined in Section 2.3. All years will be explored. Search results will be selected and exported to a Microsoft Excel spreadsheet and Endnote X9 (reference management software).

### 2.3. Searching for Articles and Comprehensiveness of the Searches

Article comprehensiveness is obtained from the search engines through inclusion and exclusion criteria [26]. Comprehensiveness includes the level of readability and ethicality. From here, the scope of the search is run first in the protocol registry to ensure that the scope of the investigations does not replicate research that has been completed or that has undergone a review process [26]. Early protocol registration provides transparency in the research process and safeguards against duplication issues. It also confirms a team’s action plan, research question, eligibility criteria, intervention/exposure, quality evaluation, and pre-analysis strategy.

The scoping exercise will use the quality assessment suggestions by Cochrane, where individual study quality will be categorized into three parts: low- (criteria fully met), moderate- (criteria partially met), and high-risk of bias (criteria not met), the requirements for which are listed in the section describing inclusion and exclusion [27,28,29,30].

### 2.4. Search Terms

The search technique uses a three-stage protocol to find only published publications (Figure 1). A preliminary search for a similar review will be conducted through the literature review database (Figure 2), which will be followed by an examination of the text words found in the titles and abstracts and the index keywords used to describe each article (keywords strategy as in Table 1). The search strategies consist of searchable keywords in the Boolean operator. Each keyword in the P, I, C, and O items will be linked with the “OR” Boolean operator, and all of the PICO items will be connected with “AND”. Special Boolean characters such as “*” will be used as inclusion criteria for possible adjectives, and “$” or “-” will be used to link each character as one keyword.

A second search will be conducted across all databases utilizing all specified keywords and index phrases. Additional research will be found in the third phase by searching the reference lists of influential papers. The studies will be limited to those written in English and published between 2019 and 2022. Medline, Science Direct, Scopus, Web of Science, the Cochrane Library of Systematic Reviews, PROSPERO Systematic Reviews on Health Science, ProQuest, Wiley, and Highwire Press will be searched.

### 2.5. Eligibility Criteria

Eligibility criteria are essential in undergoing literature selection to produce an excellent systematic review and meta-analysis. The eligibility for literature to be selected consists of inclusion and exclusion criteria to ensure the quality of the final evaluation. The quality assessment will follow the Cochrane Risk of Bias tool (Cochrane RoB2) [22,25,27]. The inclusion and exclusion criteria are as mentioned below.

#### 2.5.1. Inclusion Criteria

Our study will include research published in the allocated time frame of 2019 to 2022. Due to the emergence of COVID-19 in December 2019, the search results from 2019 to 2022 intend to capture any early reports of coronavirus studies before the virus outbreak. The 2022 search results will include all related literature that mentions all of the information relevant to this research. All of the related search criteria in the PRISMA methodologies that have the PICO statement (Population, Intervention, Comparison and Outcome) are listed in Table 1 [19]. However, Comparison keywords are not included, as there is a non-related comparison, as we include all economic and population factors in this research. The statement of the PICO keywords is an essential strategy for mining the MeSH Database (Medical Subject Headings Database). Databases that use MeSH terms include (but are not limited to) all related journals in this database (PubMed, CINAHL Complete, Medline) and are used as a criterion for selecting reputable publications to be used as references while conducting data extraction and analysis.

#### 2.5.2. Exclusion Criteria

As this protocol aims to establish a standard practice for reviews and meta-analyses, several factors would be considered exclusion criteria when selecting a publication. A publication is only regarded as reputable if it has references. Therefore, only peer-reviewed publications from Medline, Science Direct, Scopus, Web of Science, Cochrane Library of Systematic Reviews, ProQuest, Wiley, and Highwire Press will be included, and other related publications will be excluded. Non-peer reviewed literature, known as grey literature, will be excluded because (a) there are no consistent means to assess the scientific rigor of the publications and (b) there is no systematic method for retrieving this literature. The searches will be conducted in English, and only English-language literature will be included due to the project scope, timeline, and funding.

This research will also exclude patients who did not exhibit or suffer multi- or co-morbidities as pre-existing conditions before contacting COVID-19. Therefore, any complications post-contacting COVID-19 will be excluded from this research.

The most important factors are that the studies conducted in English are preferred due to the sheer number of publications to be filtered. This literature filtering is important to help reduce the cost of translating research articles and the probability of wasting resources post-translation after the elimination of unrelated themes.

However, there are no restrictions for research geography, population, type of research, and study design, as these have been deemed unrelated to this particular study.

### 2.6. Article Screening

All articles identified using the search string provided in Section 2.3 will be uploaded in EndNote X9 (reference management software), where the authors will delete any duplicate identifiers. After the duplicates have been removed, screening for the review will be evaluated based on (i) title and abstract eligibility and (ii) full-text eligibility. Screening will be assessed using the eligibility criteria listed in Section 2.4. Articles that pass the inclusion criteria and papers that seem unclear on their relevance at the title and abstract levels will be included and reviewed at the full-text level. A record of the included/excluded articles will be made for each stage. Articles excluded in the full-text step will be recorded with the reasons for exclusion as an Appendix A.

Two reviewers will conduct this screening. Kappa tests will be performed to ensure the consistency and accuracy of the decisions by the two reviewers throughout the screening process. Overall, only 10% of the literature retrieved will be selected by non-reviewer experts in the research field search and screened independently by each reviewer. The selection of 10% by a non-reviewer expert is crucial to reduce risk of bias. Consistency in decisions will be analyzed using Cohen’s Kappa test. A Kappa value of K > 0.60 is deemed significant for consistent and accurate findings [22,25,27]. In addition, any disagreements between reviewers will be discussed and resolved regardless of the Kappa value results. Suppose a retrieved publication was authored or co-authored by any of the two reviewers selected. In this case, the publication will be referred to another reviewer in the team for assessment.

### 2.7. Study Quality Assessment

A critical appraisal of study quality and strength will be carried out using standard quality assessment criteria for evaluating quantitative research adapted from critical quality appraisal assessment [25,28]. The checklist has eight (8) assessment criteria: (1) research objectives; (2) study designs; (3) study outcomes; (4) sample size calculations; (5) findings analysis; (6) variance estimations for the significant results; (7) good reported results; and (8) conclusions. The article scores two points if it satisfies the underlying criteria, one point if the article partially meets the requirements, and zero if the report does not meet any of the requirements.

### 2.8. Data Coding Strategy and Potential Effect Modifiers/Reasons for Heterogeneity

Descriptive analysis coding consists of multiple pieces of information from the articles, such as the title, article citation, year of publication, and location of the country of the lead author’s affiliated institution. Then, a narrative synthesis is carried out using a thematic category that revolves around each article’s key findings. These mainly include the study hypotheses, explaining the impact of COVID-19 on the micro-costs and economics for treating patients with pre-existing conditions. The synthesis followed a two-step coding process. First, a line-by-line review will be conducted on the articles to identify the critical impacts of alien invasive species. Each paper will be assigned a code label to cluster all of the information into a common theme. Two of the reviewers will independently perform the coding process; a discussion will be carried out with all reviewers to create a basis for a robust set of common themes.

### 2.9. Data Synthesis and Presentation

#### 2.9.1. Thematic Synthesis

Thematic synthesis will focus on the direct medical costs for COVID-19 management among patients with multi/co-morbidities and factors associated with the cost variances.

#### 2.9.2. Meta-Analysis

Because of the various outcomes measured and the sparse publication in this area, we believe that meta-analysis will be limited. Where studies have employed the same intervention- and outcome-measure type, the mixed results must be sorted according to manner. For analysis, we will utilize R Studio software for a random-effects meta-analysis to pool the data from randomized controlled trials, including standardized mean differences for continuous outcomes and risk ratios for binary outcomes, and to produce 95% confidence intervals and two-sided *p* values for each result. We will modify the standard deviations for the design impact in studies where the effects of clustering have not been considered using intraclass coefficients from the study reports or external estimates from similar studies.

The final report of the manuscript produced using this protocol will present a definition of multi- or co-morbidity status in COVID-19 patients and its derivative medical classification (for existing preconditions before contracting COVID-19), general population and sample size, risk factors, and the output of each study. The conglomerations of data will be represented in tables, and the output analysis will be discussed in detail to pinpoint research gaps for future studies. After data synthesis and categorizing the studies mentioned above, the final report will be written using the Preferred Reporting Items for Systematic Reviews and Meta-Analyses (PRISMA) standards. In this step, a webinar will be held to share the list of definitions with an international group of specialists in the field of healthcare economics and allied medical practitioners. Their opinions on the best way to define the economic impact of multi- or co-morbidities in COVID-19 patients on the healthcare system will be gathered. A final consensus will be reached utilizing Delphi methodologies.

## 3. Discussion

The ongoing COVID-19 pandemic has been associated with considerable healthcare and humanistic burden. It is expected to lead to significant economic and financial implications and clinical responsibility. Countries need to be prepared to cope with the additional pressure on the healthcare system. There are many uncertainties in COVID-19 disease, and the magnitude of the healthcare and economic consequences is unknown. The COVID-19 pandemic will impact healthcare systems and patients in many aspects. The most immediate need is to address the increased healthcare burden and the demand for healthcare resources, including healthcare personnel, healthcare facilities, and an increased load of clinical episodes. The economic implications of managing COVID-19 patients with underlying co-morbidities or multi-morbidities are huge, as those patients are usually susceptible to developing the late stage of the disease, which requires a considerable number of resources, including expensive drugs and intubation, admission to intensive care units, and numerous blood and radiological investigations.

Nevertheless, shreds of evidence on the magnitude of the economic burden of treating these patients are sparse. Therefore, the systematic review and meta-analysis that will be conducted are essential to facilitate decisions in optimizing healthcare resource use and to help divert funds from other less critical services during this crisis. During the pandemic, many difficult decisions will need to be made, and data on the economic impact due to COVID-19 can help inform national policy and decision-making.

### 3.1. Limitation of Strength

The limitations of the systematic review protocol are that the study only considers scholarly articles reflected in Medline, Science Direct, Scopus, Web of Science, Cochrane Library of Systematic Reviews, ProQuest, Wiley and Highwire Press and does not include other web databases and grey literature and irregulated literature. The language inclusion criteria include only English and no other languages. To the author’s knowledge, the strength of the protocol is that no review fills these gaps in the systematic mapping review literature, as it performs a broad search to review the costs associated with multi-morbidity treatment for SARS-CoV-2 patients.

### 3.2. Limitation of Biasness

While considering the selection strategies, there are three types of bias that need to be addressed in this manuscript: (1) selection bias, (2) information bias, and (3) confounding bias. Selection bias occurs due to the selection of only English-language manuscripts from peer-reviewed journals, which will result in a confined selection of information that results in selection bias. However, due to the reviewer’s limitation of understanding articles written in other languages, the language barrier will further result in missing data or the gathering of misinformed data and will produce misinformation when constructing the S.L.R. The selection of peer-reviewed manuscripts also removes the likelihood of selecting false or misinformation data from grey-area publications. Despite the reviewer’s acknowledgment of possible important and groundbreaking results being excluded based on these limitations, the limitations are needed to enable the manuscript to adhere to the quality standard suggested early in the S.L.R.

Information bias occurs due to the human limitation of reviewing numerous pieces of literature to accept only quality manuscripts. This bias may be due to the selection of reviewers that are inadequate or not experts. To avoid information bias, we have several reviewers on the team. The team members consist of medical doctors, public health physicians, and health economists who are experts in the field. Most of the team members are members and important stakeholders in the COVID-19 taskforce team at local and international levels.

Confounding biases are possible and are very common in health economics studies. Many factors contribute to the higher direct medical costs of patient management other than co-morbidities. For example, the hospital system, healthcare finance, clinical pathways, length of stay, age, and other factors may also contribute to cost increases. However, based on our early desk research, many studies highlight that co-morbidities are the main cost driver for the direct medical costs in COVID-19 around the world [13,14,15,18,29,30,31,32].

## 4. Conclusions

The final systematic review of the costs associated with multi-morbidity treatment for SARS-CoV-2 patients will include a detailed summary of evidence in the form of figures and tables of the study characteristics for evidence-based identification. The output from the systematic review will uncover the gap between research that justifies the economic perspectival view of COVID-19 issues in medical fields.

## 5. Patents

This research will be registered in both the PROSPERO/COCHRANE database and the *IJERPH* database as intellectual property under the Creative Commons Attribution (CC BY) license. Therefore, this research article was obligated to adhere to Creative Commons Attribution (CC BY) license conduct and its regulation.

## Figures and Tables

**Figure 1 ijerph-19-13157-f001:**
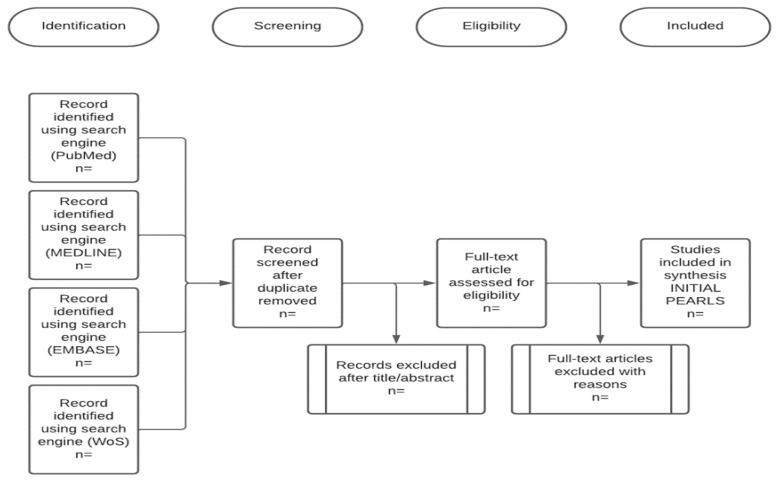
A schematic of the process of the systemic review. (Randomized controlled trials (R.C.T.s); Preferred Reporting Items for Systematic Reviews and Meta-Analyses (PRISMA).)

**Figure 2 ijerph-19-13157-f002:**
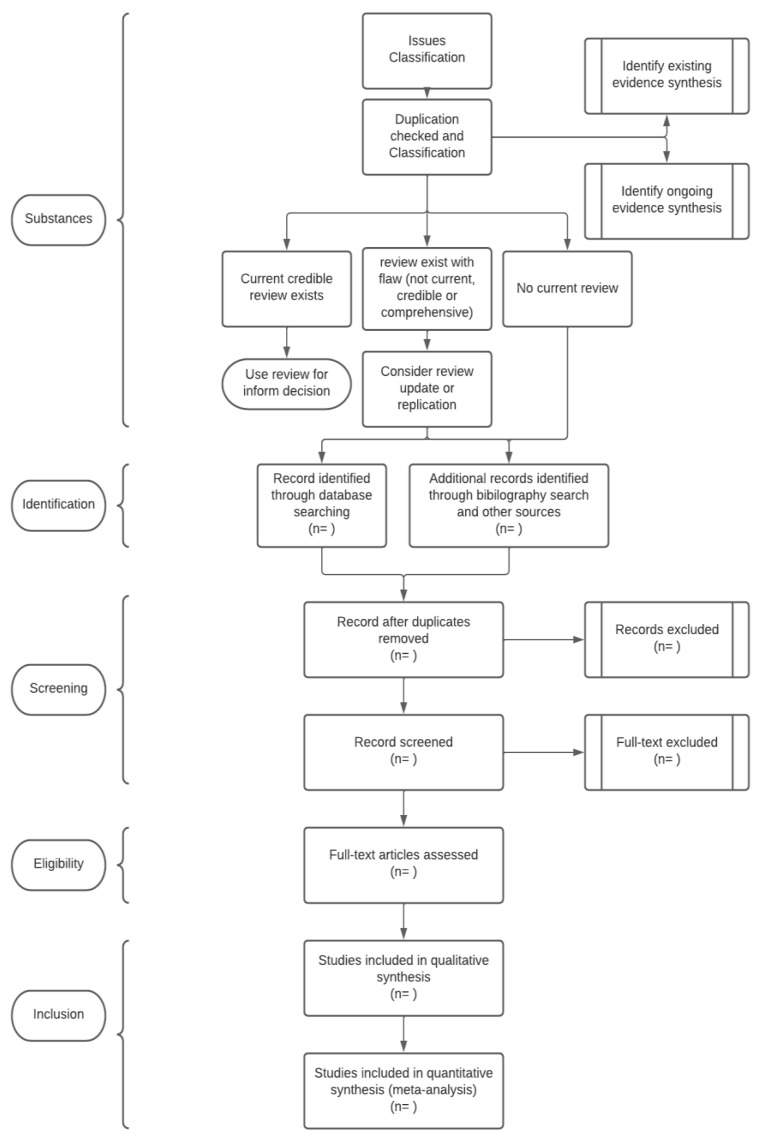
Full flow for search criteria of the systemic review. (Includes database search strategies to include or exclude specific studies from duplicate titles or subjects).

**Table 1 ijerph-19-13157-t001:** Components of the primary research questions with search strategies [17].

Item	Keywords	Search Terms	Search Strategies
P = Population/Patient/Problem	Patients suffer COVID-19	COVID-19	“COVID” OR “HCoV-19” OR “2019-nCoV” OR “SARS-CoV-2” OR “SARS-nCoV-2” OR “COVID$19*” OR (coronavirus) OR (“novel$corona*”)
I = Intervention	Patients suffer COVID-19 with multimorbidity, co-morbidity cases, AND related diseases	Multi-, co-morbidity, and related diseases	((“Multi-morbidity*”) OR (“Comorbidity*”) OR (“Multi-chronic condition”) OR (“Multiple chronic condition”) OR (“disorder*”) OR (“Multi$disorder”) OR (“dignos*”) OR (“co$dignos*”) OR (“cancer*”) OR (“cerebrovasc*”) OR (“chronic$lung*”) OR (“chronic$liver*”) OR (“cyst*”) OR (“diabetes*”) OR (“heart*”) OR (“neuro*”) OR (“immuno*”) OR (“tuberculo*”))
O = Outcome	Treatment cost for patient or world economic implication	Cost	((econom*) OR (cost*) OR (fee*) OR (“fees”) OR (charge*) OR (“organized$financing”) OR (grant*) OR (financ*) OR (reimburse*) OR (“prospective$payment*”) OR (“prospective$payment&system*”) OR (“prospective$reimbursement*”) OR (“prospective$reimbursement system*”) OR (“prospective$price*”) OR (“block$fund*”) OR (blockfund*) OR (“bulk$fund*”) OR (bulkfund*) OR (“lump$sum*”) OR (lumpsum*) OR (pay) OR (“payment”) OR (payments) OR (paying) OR (purchase) OR (purchasing) OR (purchased) OR (“price”) OR (“pricing”) OR (“fund*”) OR (capitation) OR (“regulation”) OR (“incentiv*”) OR (“econ*$implication”) OR (“econ*$burden*”) OR (“cost$of$ill*”) OR (“burden$of$ill*”) OR (“cost$of$sick*”) OR (“disease$cost*”) OR (“sick*$cost*”))

These search strategies include population, intervention, and outcome only.

## Data Availability

Not applicable.

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
