# Peer review of "Economic Burden of SARS-CoV-2 Patients with Multi-Morbidity: A Systematic Review Protocol"

_ijerph, 2022, doi:10.3390/ijerph192013157_

Round 1

Reviewer 1 Report (Previous Reviewer 3)

The protocol was modified and should be of interest to audience in the related fields. The English has also been improved.

The authors have fulfilled my concerns. I do not have any other questions.

Author Response

We would like to convey our deepest thanks to the reviewer for their hard work, constructing comments and suggestions and for pointing out the mistakes we had in the manuscript. 

Journal Article Correction Form

Manuscript Title

:

Economic Burden of SARS-nCoV-2 patients with multimorbidity: A systematic Review Protocol

Reviewer

:

Reviewer 1

Review Date

:

16 Aug 2022

Correction Date

:

22 Sept 2022

No

Reviewer Comment

Actions

1

The protocol was modified and should be of interest to audience in the related fields. The English has also been improved.

The authors have fulfilled my concerns. I do not have any other questions.

We would like to convey our deepest thanks to the reviewer for their hard work, constructing comments and suggestions and for pointing out the mistakes we had in the manuscript.

Reviewer 2 Report (Previous Reviewer 4)

I believe the authors have thoroughly and satisfactorily responded to my critiques from the first review. 

Author Response

We would like to convey our deepest thanks to the reviewer for their hard work, constructing comments and suggestions and for pointing out the mistakes we had in the manuscript. 

Journal Article Correction Form

Manuscript Title

:

Economic Burden of SARS-nCoV-2 patients with multimorbidity: A systematic Review Protocol

Reviewer

:

Reviewer 1

Review Date

:

12 Aug 2022

Correction Date

:

22 Sept 2022

No

Reviewer Comment

Actions

1

I believe the authors have thoroughly and satisfactorily responded to my critiques from the first review.

We would like to convey our deepest thanks to the reviewer for their hard work, constructing comments and suggestions and for pointing out the mistakes we had in the manuscript.

Reviewer 3 Report (Previous Reviewer 1)

My sincere congratulations to the authors for both defending their work and improving their manuscript.

* Dear authors, it is useless that you upload the PRISMA checklists if you do not complete them. Thank you very much.

Some additional comments:

·      In the Abstract, PRISMA is a Statement, not a guideline. And I think when you write “these studies”, you meant “this study”.

·      Introduction. Please provide an explanation the first time an acronym appears in the text and in the abstract. Example G.D.P.

·      Table 1. It is strange that a table is called up far away from the text and in a different section. The first figure to appear in the text is figure 2, and again it is in a separate and strange place. Check this throughout the manuscript.

·      You should provide a subtitle to the first part of the method.

·      “This systematic review protocol applies the Guideline and Standards for Cochrane Review under the COVID-END initiative by PROSPERO”. Are you sure this sentence is clear enough?

·      The section methods is still a bit chaotic. The subsection entitled Bibliographic Databases contains information on eligibility criteria, acronyms are explained and data on search strategy. It is difficult to understand. Section 2.2 is very unspecific. Please, simplify the legibility criteria.

Author Response

We would like to convey our deepest thanks to the reviewer for their hard work, constructing comments and suggestions and for pointing out the mistakes we had in the manuscript.

Journal Article Correction Form

Manuscript Title

:

Economic Burden of SARS-nCoV-2 patients with multimorbidity: A systematic Review Protocol

Reviewer

:

Reviewer 3

Review Date

:

03 Sep 2022

Correction Date

:

22 Sept 2022

No

Reviewer Comment

Actions

1

The article is fine, but it can be improved.

. My sincere congratulations to the authors for both defending their work and improving their manuscript.

We would like to convey our deepest thanks to the reviewer for their hard work, constructing comments and suggestions and for pointing out the mistakes we had in the manuscript.

* Dear authors, it is useless that you upload the PRISMA checklists if you do not complete them. Thank you very much.

Appendix

PRISMA checklist updated and filled as suggested.

Some additional comments:

·      In the Abstract, PRISMA is a Statement, not a guideline. And I think when you write “these studies”, you meant “this study”.

 Page 1, Line 36,

These study changed to this study.

·      Introduction. Please provide an explanation the first time an acronym appears in the text and in the abstract. Example G.D.P.

Page 1, Line 42,

GDP acronym full text updated.

·      Table 1. It is strange that a table is called up far away from the text and in a different section. The first figure to appear in the text is figure 2, and again it is in a separate and strange place. Check this throughout the manuscript.

Table 1 is placed under subsection 1.2 and subsequent figure 1 and figure 2 accordingly, however, the reference to table 1 in the introduction is removed to avoid confusion.

·      You should provide a subtitle to the first part of the method.

Page 2, Line 45
Subtitle 2.1. A methodological overview was added in the first part of the method. The following subtitle is numbered accordingly.

·      “This systematic review protocol applies the Guideline and Standards for Cochrane Review under the COVID-END initiative by PROSPERO”. Are you sure this sentence is clear enough?

The sentences indicate that the Prospero special registration for Covid 19 initiative for health science and medical review. We believe that it is sufficient.

·      The section methods is still a bit chaotic. The subsection entitled Bibliographic Databases contains information on eligibility criteria, acronyms are explained and data on search strategy. It is difficult to understand. Section 2.2 is very unspecific. Please, simplify the legibility criteria.

Page 3, Line 15 – 19

The Bibliographic database section was updated as recommended. The excluding criteria were removed and reinsert at exclusion criteria to avoid misinformation.

Reviewer 4 Report (New Reviewer)

The article is fine, but it can be improved.

To improve the article, authors should expand on the results and conclusions. The conclusions are very short and must be expanded.

Author Response

We would like to convey our deepest thanks to the reviewer for their hard work, constructing comments and suggestions and for pointing out the mistakes we had in the manuscript.

Journal Article Correction Form

Manuscript Title

:

Economic Burden of SARS-nCoV-2 patients with multimorbidity: A systematic Review Protocol

Reviewer

:

Reviewer 4

Review Date

:

03 Sep 2022

Correction Date

:

22 Sept 2022

No

Reviewer Comment

Actions

1

The article is fine, but it can be improved.

.

We would like to convey our deepest thanks to the reviewer for their hard work, constructing comments and suggestions and for pointing out the mistakes we had in the manuscript.

2

To improve the article, authors should expand on the results and conclusions. The conclusions are very short and must be expanded

The conclusion were short because this article focusing on the protocol of systematic review. Thus the full result and conclusion will be available in the full SLR with meta analysis after the protocol accepted and the team finish summarized all related articles according to their thematic and meta-classification.

Round 2

Reviewer 3 Report (Previous Reviewer 1)

Many thanks to the authors for answering all my questions.

This manuscript is a resubmission of an earlier submission. The following is a list of the peer review reports and author responses from that submission.

Round 1

Reviewer 1 Report

Thank you for allowing me to read this paper. The suggestions given are intended to improve your work. The same feedback will be given to editors and authors. 

Major comments:

·    If your research protocol is already published in PROSPERO, what scientific utility exists in republishing it in a journal?

·     Although the authors claim to have followed the PRISMA statement, there are no references, so I cannot know if they have used the last update. Please upload the PRISMA CHECKLIST for review protocols (PRISMA-P) from http://www.prisma-statement.org/Extensions/Protocolsnot the PRISMA-NMA as used.

·   "Because of the variety of outcomes measured and the sparse publication on this area, we believe that meta-analysis will be limited." In other words, are you trying to publish a paper with information from a protocol already registered in PROSPERO for a meta-analysis that you initially believe you will not be able to perform? Perhaps I have not understood you correctly.

Additional comments:

·      There are errors in the English language.

·  The abstract cannot exceed 200 words, according to the journal's instructions.

·      "Systematic Literature" is what you want to use as a keyword?

·      References should be joined together; this is wrong [1], [2]. ïƒ  [1,2].

·      The introduction is sparse.

·      The search string is part of the method, not part of the introduction.

·      You don't need to explain what PROSPERO is.

·      This is not a question: "The primary question for this systematic review is "to determine the direct medical costs associated in clinical management of Covid-19 patients with background comorbidities or multimorbidity", this is an aim.

·  Check the list of references to ensure that it matches the journal's requirements.

·      Section 2.2 appears to be information that has already been explained.

·      I recommend that the authors read other systematic reviews to construct the method section better.

·      Simplify eligibility criteria.

·   "The critical appraisal of study quality and strength will be carried out using standard quality assessment criteria for evaluating quantitative research adapted from" From what?

·      What about the risk of bias?

·      I don't understand the information in the patent section.

Author Response

We would like to thank the reviewer for the comment and helpful suggestion in order to complete this manuscript. 

Reviewer 2 Report

Article is well written and explained but, there is no novelty in this protocol. All the available methods are mainly elaborated rather than focusing any new idea (if any ) required for this systematic review. 

Author Response

We thank the reviewer for the suggestion and comment to correct our manuscript. 

Reviewer 3 Report

This is a systematic review protocol that guides research on the associations and differences between economic factors and the treatment of patients with COVID-19 comorbidities. The results will justify the economic view of the COVID-19 issue in the medical field and help healthcare stakeholders to further understand and inform decision-making processes.

1.    Please revise the format of Table 1. The space between the continuation part of intervention and outcome is too narrow, which is difficult to distinguish between the index keywords.

2.    More index keywords should be added to the search strategies column in Table 1, for example, besides diseases of various organs, diseases of the endocrine system including thyroid diseases, diseases of adrenal function, etc.

3.    In line 102, since the study is related to SARS-nCoV-2, why the initial screening of the literature included all the time period? It should be later than year 2018. Please limit the scope of the initial screening in order to make the work more efficient.

4.    In the protocol writing and registration part, please provide the references of systematic reviews of peer-researched COVID-19 co-morbidity and financial relevance if have.

5.    In line 145, why the search scope of these significant papers is limited to 2000-2015?

6.   in line 188, the sentence " This is to ensure other SARS-nCoV related disease to be included in these studies." How to ensure the relevance of this disease to the virus and how to confirm the supplements without a definite COVID-19 diagnosis as a screening basis?

7.    In line 206, it is recommended that non-reviewer to select the 10% of the articles for reviewer in the screening stage, so as to reduce the bias brought by subjectivity.

8.    The outcome through the protocol should be describe more specifically. For example, characteristics of included studies should be showed intuitively in a figure. PMID: 33789839 could be cited.

9.    The first part of the meta-analysis is not clear and a heterogeneity analysis should be performed before elaborating on the model. In line 247, there is a period missing.

Author Response

We would like to thank the reviewer for the comment and suggestion in order to repair this manuscript. 

Reviewer 4 Report

Overall this is a well written and clear protocol for conducting a systematic review on resource utilization among COVID patients with multimorbidity. The authors registered the study in PROSPERO which is also a positive. I have a few comments and suggestions: S

1) I think you need to add a few more terms to the COVID-19 search:

"SARS-CoV-2" which is the most common abbreviation in the U.S (the authors use "SARS-nCoV-2" which may miss the above). 

"2019-nCov" which was one of the earliest abbreviation used.

"HCov-19" also an early abbreviation

"COVID*" (The authors have COVID$19, but other variations may appear and sometimes COVID will be alone)

2) Sometimes the authors use SARS-Cov-2 and sometimes SARS-nCov-2. I think they should pick one for consistency. I believe SARS-CoV-2 is more common.

3) Some references appear duplicated. For example 1 and 5, 2 and 4, and 25 and 26. 

Author Response

We would like to thank the reviewer for their suggestion and comment. The correction was done in line with the reviewer's suggestion and comment. 

Round 2

Reviewer 1 Report

My sincere congratulations to the authors for both defending their work and improving their manuscript.

Some additional comments:

·      In the Abstract, this sentence is placed in a strange place: “These studies were registered 33 in the PROSPERO registry (ID: CRD42022323071)”.

·      It is not necessary to create heading 1.1. It should be placed at the end of the introduction. That is enough.

·      Nor do I see any sense in section 1.1.1.

·      The [18] reference is placed in a strange place.

·      Methods section:

o   I think there is still mixed information in the methodology. Under the first heading, which is called Databases, there is information on the search and on the eligibility criteria. In the second, which is supposed to be about the comprehensibility of the search, there is information about the terms. 

o   it is not clear to me the tool to see the quality of the studies, are the capital letters missing?

o   When you mention PROSPERO and COCHRANE, you should also include the PRISMA declaration.

o   There is information on the PRISMA statement and how the report will be written in the meta-analysis section.

o   It is still not clear to me how they are going to analyse the risk of bias, as the PRISMA template they have uploaded is not filled in.

·      I am not sure whether section 3.2 is relevant. Would it not be better to include information on bias risk assessment in the methodology and mention this aspect more briefly in the discussion?

·      Bibliographic references still do not comply with the journal's guidelines.

Author Response

We would like to thank the reviewer for commenting on this manuscript. The changes have been made accordingly. Major changes especially for reference citation in the document according to MDPI style in Endnote compared to IEEE citation using Mendeley. Other corrections are mentioned in the author correction form attached below. The risk of biasness and PRISMA declaration was added and updated in the articles as suggested.  
